# Generation of a Library of Carbohydrate-Active Enzymes for Plant Biomass Deconstruction

**DOI:** 10.3390/ijms23074024

**Published:** 2022-04-05

**Authors:** Vânia Cardoso, Joana L. A. Brás, Inês F. Costa, Luís M. A. Ferreira, Luís T. Gama, Renaud Vincentelli, Bernard Henrissat, Carlos M. G. A. Fontes

**Affiliations:** 1Centro de Investigação Interdisciplinar em Sanidade Animal—Faculdade de Medicina Veterinária, Universidade de Lisboa, Pólo Universitário do Alto da Ajuda, Avenida da Universidade Técnica, 1300-477 Lisboa, Portugal; luis.ferreira@reitoria.ulisboa.pt (L.M.A.F.); ltgama@fmv.ulisboa.pt (L.T.G.); 2NZYTech Ltd., Estrada do Paço do Lumiar, Campus do Lumiar, 1649-038 Lisboa, Portugal; joana.bras@nzytech.com (J.L.A.B.); ines.costa@nzytech.com (I.F.C.); 3Centre National de la Recherche Scientifique, Unité Mixte de Recherche 7257, Université Aix-Marseille, 13288 Marseille, France; renaud.vincentelli@univ-amu.fr (R.V.); bernard.henrissat@gmail.com (B.H.); 4Institut National de la Recherche Agronomique, Unité sous Contrat 1408 Architecture et Fonction des Macromolécules Biologiques, 13288 Marseille, France; 5Department of Biological Sciences, King Abdulaziz University, Jeddah 21589, Saudi Arabia

**Keywords:** carbohydrate-active enzymes (CAZymes), plant biomass, gene synthesis, PCR, high-throughput (HTP) cloning, HTP expression

## Abstract

In nature, the deconstruction of plant carbohydrates is carried out by carbohydrate-active enzymes (CAZymes). A high-throughput (HTP) strategy was used to isolate and clone 1476 genes obtained from a diverse library of recombinant CAZymes covering a variety of sequence-based families, enzyme classes, and source organisms. All genes were successfully isolated by either PCR (61%) or gene synthesis (GS) (39%) and were subsequently cloned into *Escherichia coli* expression vectors. Most proteins (79%) were obtained at a good yield during recombinant expression. A significantly lower number (*p* < 0.01) of proteins from eukaryotic (57.7%) and archaeal (53.3%) origin were soluble compared to bacteria (79.7%). Genes obtained by GS gave a significantly lower number (*p* = 0.04) of soluble proteins while the green fluorescent protein tag improved protein solubility (*p* = 0.05). Finally, a relationship between the amino acid composition and protein solubility was observed. Thus, a lower percentage of non-polar and higher percentage of negatively charged amino acids in a protein may be a good predictor for higher protein solubility in *E. coli*. The HTP approach presented here is a powerful tool for producing recombinant CAZymes that can be used for future studies of plant cell wall degradation. Successful production and expression of soluble recombinant proteins at a high rate opens new possibilities for the high-throughput production of targets from limitless sources.

## 1. Introduction

Plant biomass is the most abundant renewable source of organic carbon on Earth. The plant cell wall (PCW), its major constituent, is a highly heterogeneous and complex macromolecular structure that surrounds and protects the cell, thus playing a central role in plant survival [1]. PCW is composed of different types of recalcitrant polysaccharides, notably cellulose (40–50%), hemicellulose (25–35%), and lignin (15–20%) [2]. Microbial degradation of plant structural carbohydrates to generate usable sugars is a key step in the carbon cycle. It is of considerable biotechnological importance, particularly for the conversion of lignocellulosic biomass to biofuels, a vital objective for today’s society [3]. Thus, understanding the biochemistry of plant cell wall deconstruction is not only of biological importance but also has growing industrial significance [4,5,6].

Deconstruction of PCW carbohydrates is mediated by a large repertoire of microbial enzymes, generally termed carbohydrate-active enzymes (CAZymes). CAZymes comprise a diverse set of enzyme classes, including glycoside hydrolases (GHs), polysaccharide lyases (PLs), carbohydrate esterases (CEs), and glycosyl transferases (GTs). While PLs, CEs, and GHs carry out the breakdown of polysaccharides, GTs are mainly involved in the formation of the glycosidic bond and thus in the biosynthesis of complex carbohydrates [7]. The classification of CAZymes in families has been continuously updated in the Carbohydrate-Active EnZymes database (CAZy; www.cazy.org, accessed on 31 January 2022) [8,9]. The CAZy database is an invaluable research tool with respect to the ever-increasing amount of genomic and metagenomic information relating to carbohydrate metabolism. Currently, the database lists 172 sequence-based families of GHs, 42 families of PLs, 19 families of CEs, 114 families of GTs, and, finally, 88 families of CBMs (as of January 2022). Recently, the CAZy database incorporated a new category of proteins, named auxiliary activities (AAs), which covers redox enzymes that act cooperatively with CAZymes. The AA category of CAZymes includes lytic polysaccharide monooxygenases (LPMOs) and ligninolytic enzymes [10]. CAZymes are classified in families according to their amino acid sequence similarities, which reflect a common structural fold and catalytic mechanism.

In general, CAZymes have evolved complex molecular architectures containing one or more catalytic domains generally linked to more than one accessory non-catalytic carbohydrate-binding module (CBM) [11,12,13]. CBMs play an important role in CAZyme function by promoting the hydrolysis of insoluble substrates by the adjoined catalytic domains. CBMs bind structural carbohydrates, bringing the associated catalytic domains into close proximity of the recalcitrant substrates and thus targeting the enzyme to their specific substrates [13,14]. Similarly to CAZymes, CBMs are also classified in families according to their primary sequence similarity [9].

It is now well established that the functional diversity of CAZymes is enormous and reflects the wide multiplicity of glycan structures found in nature. Although the CAZy database provides a solid base for carbohydrate research, the existence of a tangible library of CAZymes to promote carbohydrate bioengineering research and related applications is still lacking. Here, a high-throughput (HTP) strategy was used to produce a comprehensive library of highly diverse, soluble, and pure CAZymes with known biochemical properties and to cover the maximal number of CAZy families and Enzyme Commission numbers (EC number). This strategy was also established previously to identify novel activities in a range of CAZymes with unknown functions [15]. A bioinformatic-informed selection of characterized CAZymes was initially employed, followed by HTP gene synthesis, cloning, expression, and purification of recombinant proteins. This approach not only leads to the generation of a large library of recombinant CAZymes of significant scientific interest to study PCW deconstruction but also to their applicability in the industrial and bioprocessing sectors [16].

## 2. Results and Discussion

### 2.1. Identification and Selection of CAZymes

Initially, a bank of 1955 genes encoding biochemically characterized CAZymes was constructed. All gene sequences were analyzed using a gene alignment algorithm to exclude highly similar sequences, explore the CAZy family’s diversity, and include a broad range of EC numbers. As a result, 1476 enzymes of several classes, families, and EC numbers were selected. The list of proteins included enzymes from 412 different source organisms, predominantly from bacteria but also from the other domains: Bacteria (1435 targets), Eukaryote (26 targets), and Archaea (15 targets). The 1476 en-zymes comprised 568 GHs, 58 PLs, 510 CEs, 14 AAs, 304 CBMs, and 22 others (containing different catalytic domains) (Figure 1). Some sequences were designed to include just a single module while others were maintained in their native form and all the modules were expressed. All predicted signal peptides were removed from the target sequences.

### 2.2. Gene Production and Cloning

All 1476 genes were successfully produced by amplification from genomic DNA (PCR) or by gene synthesis (GS). In total, 61% of the genes were isolated by PCR amplification using the corresponding genomic DNA as a template and the remaining 39% of the genes was codon-optimized and synthesized. Both PCR and GS products were cloned in pHTP expression vectors, encoding an N-terminal Histidine tag for protein purification. A total of 1350 genes, including all gene-encoding enzymes from the GH, CE, PL, and AA families, and some CBMs, were cloned in pHTP1 expression vector. The remaining 126 genes, encoding for CBMs, were cloned in pHTP9 expression vector, which introduced an N-terminal GFP tag into them.

### 2.3. Protein Expression and Putative Factors Affecting Solubility

The 1476 recombinant enzymes were expressed in *E. coli* BL21(DE3) and purified through IMAC using an HTP automated workstation. The molecular integrity of recombinant proteins was evaluated by SDS-PAGE. The data revealed that, in general, the molecular weight (Mw) of the purified enzymes was highly similar to the calculated theoretical value (Figure 2). In addition, 79% of the proteins were obtained in the soluble form (>20 mg of purified protein per liter of media). Since all the 1476 enzymes had previously been biochemically characterized, a higher number of targets at higher levels was expected. However, because only 1 expression condition was tested for all 1476 targets (same expression host, growth medium, induction conditions, and purification method) and that the threshold for solubility (>20 mg/L) was high, it is reasonable to expect that a proportion of ~20% of the enzymes requires further optimization of the expression conditions or a bigger culture scale to allow recovery of sufficient protein for biochemical characterization. Nevertheless, the effect of protein origin, gene production strategy, fusion with solubility tags, protein molecular weight, and amino acid composition on protein expression and solubility was evaluated.

### 2.4. The Effect of Protein Origin on Recombinant Production

The data presented in Figure 3A revealed that a significantly lower number (*p* < 0.01) of enzymes originating from eukaryotic or archaeal domains were obtained during this study. Comparison of the recombinant protein solubility across the 3 domains of life showed that a higher number of proteins from bacterial origin, 79.7% (1143 out of 1435), were obtained compared to the eukaryotic and archaeal targets, with this percentage dropping to 57.7% (15 out of 26) and 53.3% (8 out of 15), respectively (Figure 3A). The higher production rate found for bacterial targets is not surprising considering that the expression system selected was *E. coli* [17]. For example, it is widely accepted that the presence of native disulfide bonds in eukaryotic proteins represents an obstacle for proper protein folding and soluble expression in *E. coli* [17,18], resulting in the formation of insoluble inclusion bodies [19]. In addition, archaeal targets are usually poorly expressed in *E. coli* because of their origin in extreme environments and differences in host codon usage. A variety of archaeal targets originated from hyperthermophilic microorganisms, whose gene sequences usually contain a high proportion of rare codons that are poorly processed in *E. coli* [20,21]. Thus, it is expected that the expression of archaeal targets in *E. coli* results in significant levels of insoluble inclusion bodies [22].

### 2.5. Influence of the Gene Production Strategy, Gene Synthesis, or PCR on Protein Production

The effect of the gene production strategy on the yield of recombinant proteins was analyzed. The data, presented in Figure 3B, revealed that lower numbers of targets were obtained from genes synthesized artificially (*p* = 0.04). Several studies have demonstrated that codon optimization for heterologous production in *E. coli* can potentially increase recombinant protein expression, especially in sequences of eukaryotic and archaeal origin [23,24,25,26]. However, other studies revealed that *E. coli* codon optimization may lead to overexpression of recombinant proteins, thus resulting in a higher tendency to generate insoluble inclusion bodies [27,28]. The formation of inclusion bodies in recombinant expression systems occurs as a result of an imbalance between *in vivo* protein folding and aggregation [29,30,31]. The data presented here revealed that although the number of obtained proteins was reduced when genes were obtained synthetically, a significant proportion of non-viable targets revealed high levels of expression in the form of inclusion bodies (data not shown). Therefore, we can conclude that *E. coli* codon optimization increased the levels of recombinant protein expression. However, for some proteins, higher levels of expression were beyond *E. coli*’s capacity to provide proper folding and led to the accumulation of the recombinant proteins in the form of insoluble aggregates. These targets may require further testing of the expression conditions (different strains and simpler media, such as LB).

### 2.6. Does the GFP Tag Promote Protein Production?

Many studies have focused on optimizing the production processes for recombinant proteins, with the aim of reducing the accumulation of insoluble inclusion bodies. In contrast, solubilization and refolding of protein from inclusion bodies is a common strategy although it requires denaturing conditions and a subsequent renaturing step, usually resulting in poor soluble protein recoveries [32,33]. Different approaches have been developed to prevent the accumulation of inclusion bodies in *E. coli*, such as the optimization of culture conditions, co-expression with molecular chaperones [34,35], lower growth temperature during gene induction [35,36], induction expression in early-log phase culture [37], and induction with lower levels of inducer concentration, such as Isopropyl β-D-1-thiogalactopyranoside (IPTG) [38]. In addition, several fusion tags have been developed to increase the solubility of overexpressed proteins, although with variable degrees of success. Currently, available fusion systems include maltose-binding polypeptide (MBP), glutathione S-transferase (GST), ubiquitin (SUMO), thioredoxin (TrxA), and Green Fluorescent Protein (GFP) [39]. The use of a fusion partner may increase the solubility of the protein and may significantly contribute to an increase in the expression yields under different conditions. The data presented here revealed a higher number (*p* = 0.05) of proteins obtained in the soluble form when fused with GFP tag (pHTP9) (Figure 3C) compared with proteins that only included a histidine tag (pHTP1). Although this suggests higher solubility through the expression with the GFP fusion, only a controlled study in which the same proteins are expressed in the fusion with the two different fusion tags would allow a more insightful elucidation of the impact of GFP on protein solubility.

### 2.7. Influence of the Protein Molecular Weight on Protein Production

No relationship was observed between the solubility of the recombinant proteins and protein molecular weight (Figure 3D). The absence of a relationship between these two factors is not surprising. Usually, proteins with a molecular mass below 100 kDa are well tolerated in *E. coli* and are therefore expressed to considerable levels. In contrast, proteins above 100 kDa are usually not properly processed in the bacterium and hence are degraded or subjected to premature termination [39]. Other authors also suggested that protein size can affect secretion performance as large cytoplasmic proteins are physically impossible to translocate [40,41]. Since only 10 proteins in this study (Figure 3D) had a molecular mass higher than 105 kDa, it is not surprising that the protein molecular weight had no effect on production. In contrast, smaller proteins, below 10 kDa, are difficult to express stably in *E. coli* because of their improper folding and higher tendency for proteolytic degradation. These proteins can be stabilized when expressed as fusions to large proteins, such as MBP, GST, and GFP, which allow them to fold properly [39]. For the smaller CBMs that were directly cloned into pHTP9 expression vector (121 out of 304 CBMs), which contains an N-terminal GFP tag, this problem was attenuated.

### 2.8. Influence of the Primary Sequence Composition on Protein Production

Previous studies have shown a relationship between amino acid composition and recombinant protein production. Therefore, the amino acid frequency for proteins expressed in *E. coli* using the described HTP approach was analyzed. Comparison of the amino acid frequency for proteins obtained or not obtained at significant levels revealed differences in the composition of non-polar amino acids (*p* = 0.002), negatively charged amino acids (*p* < 0.001), and the content of specific amino acids (see Table 1) The non-polar amino acid group comprises glycine, cysteine, alanine, leucine, isoleucine, valine, methionine, proline, phenylalanine, and tryptophan. The negatively charged amino acid group, also named acidic amino acids, contains aspartic acid and glutamic acid. The data suggest that, in *E. coli*, a lower frequency of non-polar amino-acids and higher percentage of acidic amino acids increases the probability of a protein being soluble. This result is in agreement with Bertone et al. [42] and Christendat et al. [43], who verified that protein solubility is significantly influenced by the frequency of acidic amino acids, basic amino acids, and non-polar amino acids. Niu et al. [44] confirmed the importance of negatively charged residues. Moreover, the authors observed that dipeptides comprising acidic amino acids and basic amino acids, especially dipeptides comprising acidic amino acids, were the major determinants of protein solubility [44]. The relative content of negatively charged residues seems to be the strongest determinant of protein’s solubility as it was selected as a key attribute in several studies [42,43,45].

Analysis of the amino acid frequencies in proteins obtained with and without success revealed that lysine, tryptophan, arginine, alanine, aspartic acid, glutamic acid, glycine, and tyrosine (Table 1) are important in discriminating between insoluble and soluble proteins. In addition, the effect of aspartic acid, glutamic acid, and glutamine residues on protein solubility was previously described by Bertone et al. [42]. However, the present analysis revealed that other amino acids, in particular the relative content of lysine, tyrosine, and arginine, seem to be related to the propensity of a protein to be obtained, with a confirmed statistically significant effect (*p* < 0.002).

Over the past two decades, different studies have explored the relationship between protein solubility and amino acid sequence composition. Solubility prediction can increase the overall success rate of experiments by avoiding potentially insoluble targets and exclusively choosing promising candidates. Therefore, further analysis of the dipeptide content should be performed to develop a bioinformatics tool for CAZymes solubility prediction, based on primary sequence analysis. An accurate theoretical prediction of the solubility from a sequence is instrumental for setting priorities on targets in large-scale proteomics projects. The data presented here suggest that the predominance of specific amino acids is a good predictor of increased solubility of specific recombinant proteins. Considering the large number of CAZyme variants available in the CAZy database, we can envision that the selection of the best variants for recombinant production should consider those that present a more adequate amino acid composition for the soluble expression in *E. coli.*

### 2.9. CAZymes Activity

To confirm that the library of soluble recombinant CAZymes displayed the expected activities, a subset of the CAZYmes produced was functionally tested. Thus, all 16 xylanases (EC3.2.1.8) and 11 glucuronoyl esterases (EC 3.1.1.-) of the bank of 1166 soluble proteins were functionally tested using their target substrates, under standard conditions. The data revealed that all 27 recombinant CAZymes displayed the expected activity, thus suggesting that all enzymes of the bank are active when expressed in their recombinant form and reflecting their biochemical properties described in the literature.

The influence of enzyme activity (as expressed by the EC number) on protein solubility was evaluated. As described in Table 2, the biochemical properties of the 1476 CAZymes produced in this study covered 153 different EC number activities. The activity profile of the soluble proteins was compared with the original bank, and it was observed that 17 enzyme activities (EC numbers 1.1.3.-, 1.14.99.54, 2.4.1.140, 2.4.1.20, 2.4.1.216, 2.4.1.8, 3.2.1.109, 3.2.1.111, 3.2.1.120, 3.2.1.136, 3.2.1.157, 3.2.1.164, 3.2.1.177, 3.2.1.54, 3.2.1.61, 4.2.2.16, 4.2.2.24) were absent in the bank of the 1166 soluble enzymes. Most of these activities were represented only by one member (only two with three members) and thus the limited number of sequences per EC number limits the statistical analysis. The number of enzyme activities covered by the bank is depicted in Table 2.

### 2.10. Final Outputs

In the present study, we started with 1955 putative candidates from 486 different organisms, covering 59% and 62% of the families and EC numbers described in the CAZy database, respectively (Figure 4A,B) (www.cazy.org, accessed on 31 January 2022). Following sequence analysis, 1476 enzymes (see Appendix A), comprising approximately 55% of the families and EC numbers described in the CAZy database, were selected for production (Figure 4C). From 1166 CAZymes obtained in the soluble form (Figure 4D), 732 proteins were selected to build a large and diverse CAZyme library to develop fundamental and applied research (Figure 4E). Together with a previously implemented bank, the present library includes 1023 CAZymes (Figure 4E), belonging to 218 different organisms with different origins: 993 bacterial, 21 eukaryotic, and 9 archaeal. These enzymes are distributed in 5 CAZyme classes, comprising many families, and comprising 638 GHs, 77 PLs, 137 CEs, 9 AAs, and 162 CBMs (Figure 5). Therefore, approximately 57% of all families described in the CAZy database are covered by the present recombinant library, which constitutes a valuable resource for exploiting the biological and biotechnological potential of carbohydrates.

## 3. Materials and Methods

### 3.1. Identification and Selection of CAZymes

CAZymes were drawn from the continuously updated CAZy database, from which 1955 biochemically characterized enzymes from distinct classes, families, and activities described in the literature were identified. The diversity of EC numbers was also explored. To tune the selection, a gene alignment algorithm (NZYTech Genes & Enzymes, Lisbon, Portugal) was used to exclude highly similar sequences, namely primary sequences with homology higher than 90%. Signal peptides were predicted using SignalP 4.1 (https://services.healthtech.dtu.dk/service.php?SignalP-5.0, accessed on 31 January 2022; [46]) and removed from all candidate sequences. Enzyme domains were predicted using dbCAN, which provides pre-computed CAZyme sequence and annotation data for several bacterial genomes [47].

### 3.2. Polymerase Chain Reaction (PCR) and Gene Synthesis (GS)

Genes encoding 1476 CAZymes were obtained through PCR or GS (Appendix A). When genomic DNA was commercially available (DSMZ-German Collection of Microorganisms and Cell Cultures GmbH., Braunschweig, Germany), genes were obtained by PCR. DNA amplifications were performed with Supreme NZYProof DNA Polymerase (NZYTech Genes & Enzymes, Lisbon, Portugal) using genomic DNA as the template and gene-specific primers. The PCR conditions followed the manufacturer’s protocols. Following amplification, the assembled PCR products were purified using an NZYDNA Clean-up 96 well plate (NZYTech Genes & Enzymes, Lisbon, Portugal) in a Liquid Handling Robot workstation (TECAN, Freedom EVO series, Mannedorf, Switzerland). On the other hand, the primary sequences of CAZymes, for which no template was available, were obtained by de novo gene synthesis. Gene sequences were designed by back translating the protein sequences and optimizing codon usage for high levels of expression in *E. coli*, using the ATGenium codon optimization algorithm. In addition, genes were designed to ensure a Codon Adaptation Index (CAI) value higher than 0.8. Synthetic genes were produced in an HTP pipeline using optimized procedures, as described in Sequeira et al. [48,49]. The databank describing the 1476 genes and enzymes is presented in Appendix A.

### 3.3. High-Throughput Cloning, Transformation, and Sequencing

Purified PCR and GS products were cloned into pHTP1 or pHTP9 expression vectors (NZYTech Genes & Enzymes, Lisbon, Portugal) using NZYEasy Cloning & Expression kits I and IX, respectively (NZYTech Genes & Enzymes, Lisbon, Portugal), according to the protocol reported in Turchetto et al. [50] and Duhoo et al. [51]. Following the cloning reaction, recombinant plasmids were transformed using an HTP method into *E. coli* NZY5α competent cells (NZYTech Genes & Enzymes, Lisbon, Portugal). The transformed bacteria were spread on LB-agar kanamycin 24-deep-well plates (24-DW). After an overnight incubation at 37 °C, 1 colony per transformation was picked and grown in liquid LB media supplemented with 50 μg/mL of kanamycin in 24-DW plates (5 mL) sealed with gas-permeable adhesive seals. The plasmids were purified from the bacterial pellets using an NZYMiniprep 96-well plate kit (NZYTech Genes & Enzymes, Lisbon, Portugal) on a Tecan workstation (TECAN, Freedom EVO series, Mannedorf, Switzerland). All constructs were completely sequenced in both directions to ensure 100% consistency with the gene sequences.

### 3.4. High-Throughput Protein Expression, Purification, and Quantification

The protein expression and purification steps were based on the protocol described by Duhoo et al. [51] with few modifications. *E. coli* BL21 (DE3) cells (NZYTech Genes & Enzymes, Lisbon, Portugal) were used to transform the recombinant pHTP clones. Recombinant strains were cultured in 24-DW with 5 mL of Auto-Induction LB medium supplemented with 50 μg/mL of kanamycin and grown at 37 °C for 4 h followed by 18 h at 25 °C. At the end of the culture, cells were harvested by centrifugation at 1500× *g* for 15 min at 4 °C. The cell pellets were resuspended in 1 mL of NZY Bacterial Cell Lysis Buffer (NZYTech Genes & Enzymes, Lisbon, Portugal) with 4 μg/mL of DNAse and 100 μg/mL of lysozyme. The proteins were then purified by immobilized metal ion-affinity chromatography (IMAC) and all the steps were automated on a Tecan workstation (TECAN, Freedom EVO series, Mannedorf, Switzerland) containing a vacuum manifold. Briefly, 1 mL of crude cell lysate was incubated with 200 μL of Ni^2+^ Sepharose 6 Fast flow resin (GE Healthcare, 17-5318-02) with bound nickel and then transferred into 96-well filter plates (20 μm) (Macherey–Nagel). The wells were washed twice with 1000 μL of buffer A (50 mM NaHepes, 1 M NaCl, 10 mM imidazole, 5 mM CaCl_2_, pH 7.5) followed by 1 wash with 1000 μL of buffer B (NaHepes, 1 M NaCl, 35 mM imidazole, 5 mM CaCl_2_, pH 7.5), and, finally, the proteins were eluted with 300 µL of elution buffer (NaHepes, 1 M NaCl, 300 mM imidazole, 5 mM CaCl_2_, pH 7.5). The integrity of purified recombinant proteins, in terms of the solubility, purity, and molecular weight, was assessed by sodium dodecyl sulphate 14% polyacrylamide gel electrophoresis (SDS-PAGE) and the protein concentration was measured using the Bradford protein assay (NZYTech, Lisbon, Portugal) against a standard curve of bovine serum albumin (BSA). Recombinant proteins’ molecular weight and extinction coefficient were calculated using the ExPASy–ProtParam tool (http://www.expasy.ch/tools/protparam.html, accessed on 31 January 2022).

### 3.5. Enzyme Activity

Determination of xylanase (EC3.2.1.8) and glucuronoyl esterase (EC 3.1.1.-) activity was performed as described by Correia et al. [52] and Hüttner et al. [53], respectively.

### 3.6. Statistical Analyses

The effects of different organisms with origin, gene production strategies, vector with GFP tag, and molecular masses were analyzed with the chi-squared test. Data related to the primary sequence analysis were subjected to ANOVA according to the general linear model procedures. All statistical analyses were performed with a 95% confidence interval and conducted with SAS [54].

## 4. Conclusions

The main drawback of biomass use is related to the complexity of macromolecular polysaccharides’ composition, which requires a wide plethora of CAZymes presenting a broad spectrum of specificities. The strategy used in this study combined bioinformatic tools for enzyme selection and an optimized HTP platform for the production, cloning, expression, and purification of recombinant proteins with a wide diversity of enzyme specificities. This approach resulted in a library comprising more than 1000 CAZymes involved in carbohydrate degradation, making its application in both industrial and bioprocessing sectors possible. The data presented here reinforce the observations that different factors, such as production strategies and protein primary sequence, play a major role in determining the expression and solubility of a protein heterologously produced in *E. coli*. Further analysis of the amino acid composition of the proteins selected for this study must be performed to develop innovative strategies to predict protein solubility. Nevertheless, the effectiveness of the HTP approach described here is illustrated by the high number of soluble targets obtained, most of them with high solubility rates. Taken together, these studies reveal that the implemented strategy described here is a powerful tool not only for CAZymes production but also for the generation of large libraries of recombinant proteins that will allow exploration of the biological functions of the extensive genomic and meta-genomic information available from various sources.

## Figures and Tables

**Figure 1 ijms-23-04024-f001:**
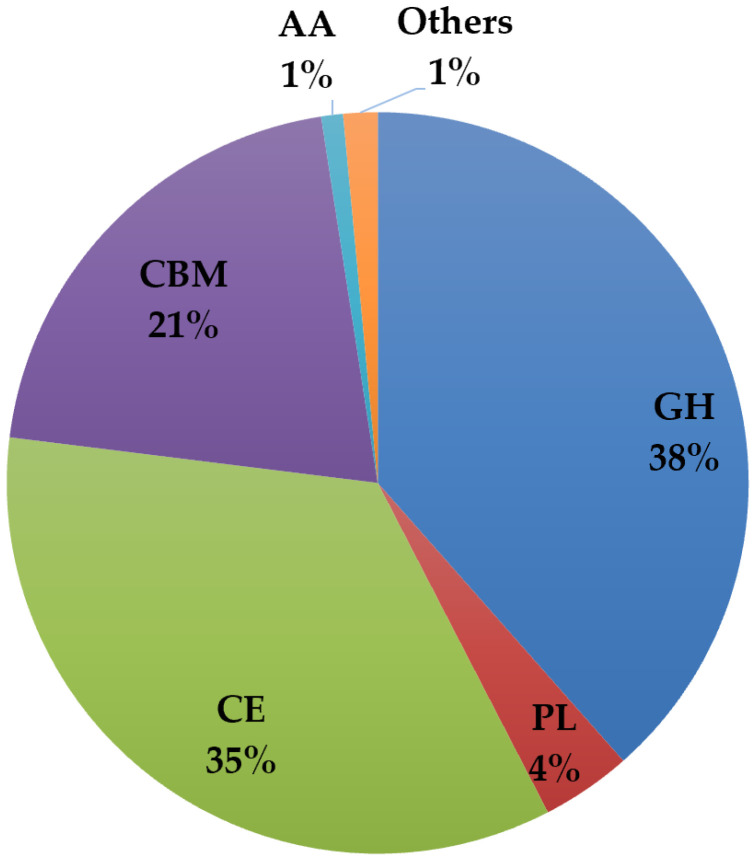
The distribution of the 1476 enzymes produced in this study by the 5 CAZy classes.

**Figure 2 ijms-23-04024-f002:**
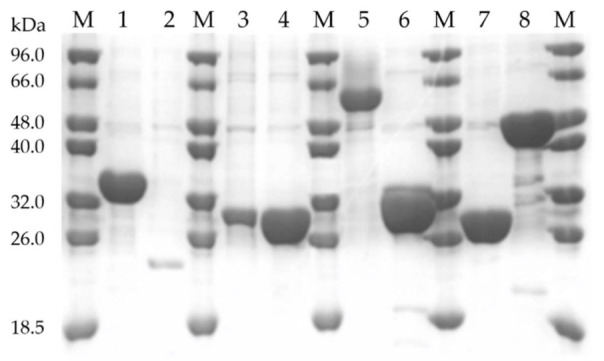
Protein expression and purity analysis of CAZymes exemplified by SDS-PAGE. Lane M contains NZYTech Low Molecular Weight (LMW) Protein Marker; Lanes 1–8 contain purified recombinant proteins with the following accession numbers: (1) ABD81807.1, (2) AAK06049.1, (3) AAG04556.1, (4) CAA84537.1, (5) CAC83072.1, (6) CAA84537.1, (7) CAB55348.1, and (8) AAP09638.1.

**Figure 3 ijms-23-04024-f003:**
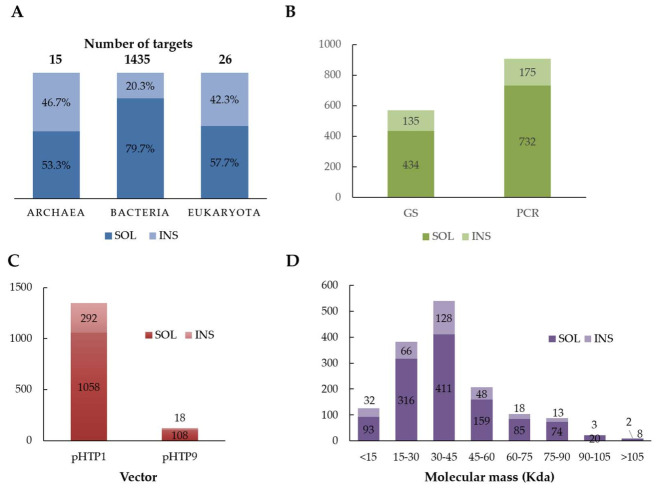
Efficacy of the recombinant production of CAZymes. (**A**) The three domains of life (proportion of recombinant proteins obtained among eukaryotic, bacteria, and archaea targets); (**B**) production strategy *(GS*, genes synthetized with codon-optimization for *E. coli*; *PCR*, genes obtained from genomic DNA); (**C**) expression vectors: pHTP1, expression vector encoding an N-terminal histidine tag; pHTP9, vector containing an N-terminal GFP tag; (**D**) protein’s molecular mass distribution. SOL: soluble protein, INS: insoluble protein.

**Figure 4 ijms-23-04024-f004:**
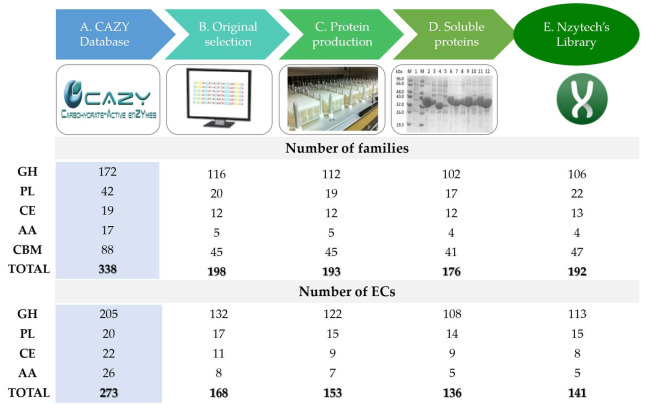
Number of CAZy families and ECs obtained in this study. (**A**) CAZy database numbers (www.cazy.org, accessed on 31 January 2022), (**B**) selected biochemically characterized CAZymes, (**C**) CAZymes produced in this study, (**D**) enzymes obtained in the soluble form, and (**E**) final Nzytech’ CAZy commercial library.

**Figure 5 ijms-23-04024-f005:**
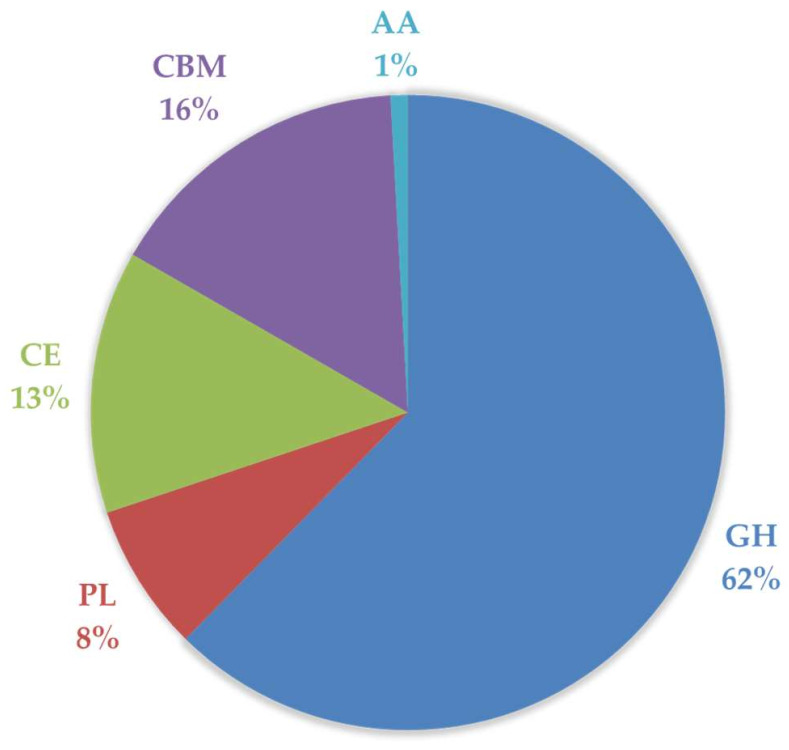
Number of CAZy classes and families represented in the current library.

**Table 1 ijms-23-04024-t001:** Analysis of the amino acid composition in primary sequences of soluble (SOL) and insoluble (INS) CAZymes produced in this study.

	INS	SOL	SEM	*p*-Value
(n)	310	1166		
Amino acid groups (%)				
Non-polar amino acids ^1^	51.4 ^a^	50.2 ^b^	0.31	0.002
Polar neutral amino acids ^2^	26.0	26.0	0.38	0.976
Negatively charged amino acids ^3^	11.2 ^b^	12.2 ^a^	0.17	0.001
Positively charged amino acids ^4^	11.4	11.5	0.17	0.420
Amino acid contents (%)				
Isoleucine (I)	5.0	5.2	0.11	0.051
Leucine (L)	7.6	7.4	0.15	0.213
Lysine (K)	4.4 ^b^	5.0 ^a^	0.15	0.001
Methionine (M)	1.9	2.0	0.06	0.138
Phenylalanine (F)	4.1	4.1	0.08	0.828
Threonine (T)	6.2	6.0	0.14	0.231
Tryptophan (W)	2.3 ^a^	2.2 ^b^	0.07	0.042
Valine (V)	6.5	6.5	0.10	0.610
Arginine (R)	4.7 ^a^	4.3 ^b^	0.12	0.005
Histidine (H)	2.3	2.3	0.07	0.659
Alanine (A)	9.0 ^a^	8.5 ^b^	0.20	0.014
Asparagine (N)	5.1	5.2	0.15	0.511
Aspartic acid (D)	6.0 ^b^	6.5 ^a^	0.10	0.001
Cysteine (C)	1.0	1.0	0.07	0.784
Glutamic acid (E)	5.2 ^b^	5.6 ^a^	0.12	0.002
Glutamine (Q)	3.6	3.4	0.09	0.287
Glycine (G)	8.9 ^a^	8.6 ^b^	0.13	0.016
Proline (P)	5.0	4.9	0.10	0.418
Serine (S)	6.6	6.5	0.15	0.246
Tyrosine (Y)	4.5 ^b^	4.9 ^a^	0.11	0.001

SEM: Standard error of the mean; Means in the same line with different letter superscripts (^a^, ^b^) are significantly different (*p* < 0.05) or tend to be significantly different (*p* < 0.1). ^1^ The non-polar amino acid group is composed of glycine (G), cysteine (C), alanine (A), leucine (L), isoleucine (I), valine (V), methionine (M), proline (P), phenylalanine (F), and tryptophan (W); ^2^ The polar neutral amino acid group is composed of asparagine (N), glutamine (Q), serine (S), threonine (T), and tyrosine (Y); ^3^ The negatively charged amino acid group, also named acidic amino acids, comprises aspartic acid (D) and glutamic acid (E); ^4^ Positively charged amino acids, also named basic amino acids, comprise arginine (R), histidine (H), and lysine (K).

**Table 2 ijms-23-04024-t002:** Activity of the 1166 soluble recombinant enzymes organized by CAZy family and expressed through the EC number.

CAZy Family	EC Number
AA1	1.10.3.2
AA3	1.1.3.12
AA7	1.1.3.-
AA10	1.-.-.-/1.14.99.54
AA NC	1.10.3.-/1.3.3.5
CE1	3.1.1.73
CE2	3.1.1.72
CE3	3.1.1.72
CE4	3.2.1.8/3.5.1.-
CE6	3.1.1.72
CE7	3.1.1.41/3.1.1.72
CE8	3.1.1.11
CE9	3.5.1.25
CE11	3.5.1.-
CE12	3.1.1.-/3.1.1.72
CE14	3.5.1.-/3.5.1.89
CE15	3.1.1.-/3.1.1.72
GH1	3.2.1.-/3.2.1.21/3.2.1.23/3.2.1.25/3.2.1.37/3.2.1.74/3.2.1.85/3.2.1.86
GH2	3.2.1.23/3.2.1.25/3.2.1.31/3.2.1.165
GH3	3.2.1.21/3.2.1.37/3.2.1.45/3.2.1.52/3.2.1.74/3.2.1.120
GH4	3.2.1.20/3.2.1.22/3.2.1.67/3.2.1.86/3.2.1.122/3.2.1.139
GH5	3.2.1.4/3.2.1.8/3.2.1.73/3.2.1.74/3.2.1.78/3.2.1.91/3.2.1.123/3.2.1.132/3.2.1.151
GH6	3.2.1.4
GH8	3.2.1.4/3.2.1.73/3.2.1.132/3.2.1.156
GH9	3.2.1.-/3.2.1.4/3.2.1.91/3.2.1.151/3.2.1.165
GH10	3.2.1.4/3.2.1.8
GH11	3.2.1.8
GH12	3.2.1.4/3.2.1.151
GH13	2.4.1.4/2.4.1.7/2.4.1.18/2.4.1.19/2.4.1.25/3.2.1.1/3.2.1.4/3.2.1.10/3.2.1.20/3.2.1.41/3.2.1.68/3.2.1.70/3.2.1.93/3.2.1.98/3.2.1.133/3.2.1.135/3.2.1.141/5.4.99.11/5.4.99.15/5.4.99.16
GH14	3.2.1.2
GH15	3.2.1.3/3.2.1.28/3.2.1.70
GH16	3.2.1.-/3.2.1.4/3.2.1.6/3.2.1.39/3.2.1.73/3.2.1.81/3.2.1.83/3.2.1.103/3.2.1.178
GH17	2.4.1.-
GH18	3.2.1.-/3.2.1.14/3.2.1.96
GH19	3.2.1.14
GH20	3.2.1.52
GH23	3.2.1.17/4.2.2.n1
GH24	3.2.1.17
GH25	3.2.1.17
GH26	3.2.1.78/3.2.1.100
GH27	3.2.1.88/3.2.1.94
GH28	3.2.1.15/3.2.1.67/3.2.1.82
GH29	3.2.1.51/3.2.1.111
GH30	3.2.1.8/3.2.1.31/3.2.1.38/3.2.1.136/3.2.1.164
GH31	3.2.1.-/3.2.1.20/3.2.1.84/2.4.1.161/3.2.1.177
GH32	3.2.1.26/3.2.1.64/3.2.1.65/3.2.1.80/3.2.1.153/4.2.2.16
GH33	3.2.1.-/3.2.1.18/2.4.1.-
GH35	3.2.1.23/3.2.1.165
GH36	3.2.1.22/3.2.1.49
GH37	3.2.1.28
GH38	3.2.1.-/3.2.1.24
GH39	3.2.1.37
GH42	3.2.1.-/3.2.1.23
GH43	3.2.1.-/3.2.1.37/3.2.1.55/3.2.1.99
GH44	3.2.1.4
GH45	3.2.1.4
GH46	3.2.1.132
GH47	3.2.1.113
GH48	3.2.1.4/3.2.1.176
GH49	3.2.1.11/3.2.1.95
GH50	3.2.1.23/3.2.1.81
GH51	3.2.1.55
GH52	3.2.1.37
GH53	3.2.1.89
GH55	3.2.1.39
GH57	2.4.1.18/2.4.1.25/3.2.1.1/3.2.1.41/3.2.1.54
GH62	3.2.1.55
GH63	3.2.1.20/3.2.1.84/3.2.1.170
GH64	3.2.1.39
GH65	2.4.1.8/2.4.1.64/2.4.1.216/2.4.1.230/2.4.1.279/2.4.1.282
GH66	3.2.1.11
GH67	3.2.1.139
GH68	2.4.1.9/2.4.1.10/3.2.1.26
GH70	2.4.1.5/2.4.1.140/2.4.4.-
GH73	3.2.1.- /
GH74	3.2.1.-/3.2.1.4
GH75	3.2.1.132
GH76	3.2.1.101
GH77	2.4.1.25
GH78	3.2.1.40
GH79	3.2.1.31
GH80	3.2.1.132
GH81	3.2.1.39
GH82	3.2.1.157
GH84	3.2.1.35/3.2.1.52/3.2.1.169
GH85	3.2.1.96
GH86	3.2.1.81/3.2.1.178
GH87	3.2.1.59/3.2.1.61
GH88	3.2.1.-
GH91	4.2.2.18
GH92	3.2.1.-/3.2.1.24/3.2.1.113
GH94	2.4.1.-/2.4.1.20/2.4.1.49
GH95	3.2.1.51/3.2.1.63
GH97	3.2.1.3/3.2.1.20/3.2.1.22
GH98	3.2.1.102
GH99	3.2.1.130
GH100	3.2.1.26
GH101	3.2.1.97
GH102	4.2.2.n1
GH103	4.2.2.n1
GH104	4.2.2.n1
GH105	3.2.1.-/3.2.1.172
GH106	3.2.1.40
GH107	3.2.1.-
GH108	3.2.1.17
GH109	3.2.1.49
GH110	3.2.1.-/3.2.1.22
GH111	3.2.1.-
GH112	2.4.1.211/2.4.1.247
GH113	3.2.1.78
GH114	3.2.1.109
GH115	3.2.1.-
GH116	3.2.1.21/3.2.1.37/3.2.1.52
GH117	3.2.1.-
GH118	3.2.1.81
GH119	3.2.1.1
GH120	3.2.1.37
GH121	3.2.1.-
GH122	3.2.1.20
GH123	3.2.1.53
GH125	3.2.1.-
GH126	3.2.1.-
GH127	3.2.1.185
GH129	3.2.1.49
GH130	2.4.1.281/2.4.1.319
GH134	3.2.1.78
GH137	3.2.1.31
GH142	3.2.1.185
GH143	3.2.1.185
PL1	4.2.2.2/4.2.2.10
PL2	4.2.2.2/4.2.2.9
PL3	4.2.2.2
PL4	4.2.2.-
PL5	4.2.2.3
PL6	4.2.2.-/4.2.2.3
PL7	4.2.2.-/4.2.2.3/4.2.2.11
PL8	4.2.2.1/4.2.2.5/4.2.2.12/4.2.2.20
PL9	4.2.2.2/4.2.2.9
PL10	4.2.2.2
PL11	4.2.2.23/4.2.2.24
PL12	4.2.2.8
PL13	4.2.2.7
PL15	4.2.2.-/4.2.2.3
PL17	4.2.2.-
PL18	4.2.2.3/4.2.2.11
PL21	4.2.2.7/4.2.2.8
PL22	4.2.2.6
PL24	4.2.2.-

## Data Availability

Not applicable.

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
