# Peer review of "Generation of a Library of Carbohydrate-Active Enzymes for Plant Biomass Deconstruction"

_ijms, 2022, doi:10.3390/ijms23074024_

Round 1

Reviewer 1 Report

The authors described expression of a large numbers of genes involved in plant biomass degradation. The results were summarized in statistical terms. Such a statistic study like this seems to be rare and gives us important information on effective methods of construction of expression systems. The manuscript was well-written, and I have only a few comments as described below.

section 2.8

What can be done in terms of amino acid composition to solubilize each insoluble protein? Examples or your opinions on that would be helpful to understand the importance of this section. For example, is it to get the ratios of specific amino acids closer to the soluble proteins’ ones?

L196    Others authors   --> Other authors  ??

L267    the exploit   -->   to exploit  ??

L325, L331 and so on    Lysozyme and Imidazole 

   Capital letters “L” and “I” are a style for this journal?

Table 1  There is no footnote for “SEM”.

Author Response

Responses to Reviewer 1 
Comments and Suggestions for Authors 
The authors described expression of a large numbers of genes involved in plant biomass 
degradation. The results were summarized in statistical terms. Such a statistic study like this 
seems to be rare and gives us important information on effective methods of construction of 
expression systems. The manuscript was well-written, and I have only a few comments as 
described below. 
Section 2.8
What can be done in terms of amino acid composition to solubilize each insoluble protein? 
Examples or your opinions on that would be helpful to understand the importance of this 
section. For example, is it to get the ratios of specific amino acids closer to the soluble proteins’ 
ones? 

Reply: We agree with the reviewer and have included a few sentences in the paper where we 
advance recommendations for future selection of CAZyme targets for expression in E. coli. 

Minor comments 
L196 Others authors --> Other authors ?? 
L267 the exploit --> to exploit ?? 
L325, L331 and so on lysozyme and imidazole; Capital letters “L” and “I” are a style for this 
journal? 
Table 1, There is no footnote for “SEM”. 

Reply: We have corrected all the suggestions made by the reviewer

Author Response

Responses to Reviewer 2: 
This manuscript reports the preparation of a library of carbohydrate-active enzymes for 
plant biomass deconstruction. Through the use of PCR or gene synthesis techniques, 
1476 genes encoding a diverse library of CAZymes have been cloned and expressed in 
recombinant Escherichia coli cells. Compared to CAZmes of eukaryotic and archaeal 
origin, a greater number (79.7%) of bacterial enzymes are expressed as a soluble form. 
Moreover, GFP tag can significantly improve the effectiveness of soluble expression. Despite of 
I support the publication of this work, I have a major comment and a few suggestions for the 
authors: 

Major comment 
The authors employ high-throughput strategy for cloning, expression, and purification of 
recombinant CAZymes and obtain a high number of soluble proteins. However, in my opinion, 
the biological activity of these soluble proteins is critical for their future applications in the 
deconstruction of plant biomass. Therefore, the authors are suggested to do the activity assays 
for the purified enzymes. 

Reply: We agree with the reviewer that activity of recombinant enzymes is critical under the 
scope of this project. Thus, only enzymes that were thoroughly characterized and which 
biochemical properties are described in detail in the literature were selected for the project. As 
described in literature, the great majority of these enzymes were produced recombinantly in E. 
coli ensuring that activity was previously validated for the recombinant host. Currently we are 
collecting the information of all biochemical activities of enzymes of this bank, but this will be 
part of a second manuscript. Presently the number of activities already characterized is very 
limited but confirm the assumption that all enzymes present the levels of activity reported in 
literature.

Minor comments 
Figure 2: Please provide more information about lanes 1 - 8. 
References: The scientific names of organisms should be italicized. 

Reply: We have corrected all the suggestions made by the reviewer

Round 2

Author Response

Response: We thank the suggestion that we have considered carefully. The paper aims to describe an innovative method/strategy to generate large banks of recombinant CAZymes to promote research and development. In detail, it discusses how different factors affect the solubility of the recombinant proteins, which is usually impossible when studies are restricted to few proteins. Previously, reviewer 2 suggested that the activity of the 1166 soluble enzymes to be described in the scope of the paper. In the previous response we considered 
this unappropriated for various reasons in particular: (i) the bank was selected from recombinant enzymes previously described in the literature; (ii) in the majority of cases majority of proteins were produced in E. coli. In addition, as the reviewer/editor should appreciate it is impossible in the scope of a few days to provide biochemical data concerning thousands of enzymes involving hundreds of different enzyme activities. Nevertheless, we considered very valuable the comment in terms of validating the procedures described in the paper to generate functional proteins. Thus, we have selected two groups of enzymes (xylanases and glucuronyl esterases) and proceed with the determination of enzyme activities of all enzymes expressing these activities of the bank. The results are now included in the manuscript in a new section 2.9, as suggested by the editor. The data revealed that all the 27 enzymes of the library display the expected activity. We believe this is a solid prove of concept that validates the approach described in the paper to generate large banks of functional 
enzymes. In addition, we have included a new table, Table 2, which describes all EC number activities expressed by enzymes of the bank. This will allow enzymologists, as suggested by the editor, to quickly identify the enzyme portfolio included in the bank, while more detailed information is available in Table S1. In resume, we believe the corrections made satisfy the queries raised and that the manuscript will provide a solid platform to approach plant cell wall 
research with novel innovative enzyme banks.

Round 3

Reviewer 2 Report

The raised comment has been partly addressed by the authors.